

# Methods matter: the relationship between strength and hypertrophy depends on methods of measurement and analysis

Andrew D. Vigotsky[1], Brad J. Schoenfeld[2], Christian Than[3] and J. Mark Brown[3]

[1] Department of Biomedical Engineering, Northwestern University, Evanston, IL, United States of America
[2] Department of Health Sciences, City University of New York, Herbert H. Lehman College, Bronx, NY, United States of America
[3] School of Biomedical Sciences, University of Queensland, St. Lucia, Queensland, Australia

Corresponding author
Andrew D. Vigotsky,
avigotsky@gmail.com

## ABSTRACT

**Purpose**. The relationship between changes in muscle size and strength may be affected by both measurement and statistical approaches, but their effects have not been fully considered or quantified. Therefore, the purpose of this investigation was to explore how different methods of measurement and analysis can affect inferences surrounding the relationship between hypertrophy and strength gain.

**Methods**. Data from a previous study—in which participants performed eight weeks of elbow flexor training, followed by an eight-week period of detraining—were reanalyzed using different statistical models, including standard between-subject correlations, analysis of covariance, and hierarchical linear modeling.

**Results**. The associative relationship between strength and hypertrophy is highly dependent upon both method/site of measurement and analysis; large differences in variance accounted for (VAF) by the statistical models were observed (VAF = 0–24.1%). Different sites and measurements of muscle size showed a range of correlations coefficients with one another ($r = 0.326$–$0.945$). Finally, exploratory analyses revealed moderate-to-strong relationships between within-individual strength-hypertrophy relationships and strength gained over the training period ($\rho = 0.36$–$0.55$).

**Conclusions**. Methods of measurement and analysis greatly influence the conclusions that may be drawn from a given dataset. Analyses that do not account for inter-individual differences may underestimate the relationship between hypertrophy and strength gain, and different methods of assessing muscle size will produce different results. It is suggested that robust experimental designs and analysis techniques, which control for different mechanistic sources of strength gain and inter-individual differences (e.g., muscle moment arms, muscle architecture, activation, and normalized muscle force), be employed in future investigations.

## INTRODUCTION

The combined actions of neural input, muscles, and the joint(s) about which those muscles act serve to produce sufficient endpoint force for physical function, allowing the

performance of activities of daily living, as well as the spectrum of athletic endeavors. Due to the complexity of the neuromuscular and musculoskeletal systems, many factors can influence strength, including, but not limited to, muscle moment arm, muscle size, activation, muscle architecture, and normalized muscle force (or specific tension) (*Vigotsky, Contreras & Beardsley, 2015*). Muscle size is of particular interest, as (1) it is highly plastic (*Fluck & Hoppeler, 2003*) and (2) a clear positive relationship exists between baseline muscle cross-sectional area (CSA) and strength, with greater CSAs correlating with greater strength capacities (*Maughan & Nimmo, 1984*; *Maughan, Watson & Weir, 1984*; *Schantz et al., 1983*). However, this relationship is not necessarily linear, as several additional factors interactively influence strength capacity (*Vigotsky, Contreras & Beardsley, 2015*); studying the role of and relationship between muscle size and strength is therefore less straightforward under longitudinal contexts.

While the cross-sectional correlation between muscle mass and strength remains well-established, some researchers have recently challenged the belief that resistance training (RT)-induced hypertrophy significantly impacts the ability to produce force, claiming improvements in these outcomes are separate and unrelated adaptations (*Buckner et al., 2016a*). Indeed, data remain somewhat equivocal on the relationship between changes in size and changes in strength resulting from regimented RT: A considerable range of correlation coefficients have been observed, from ∼0 to ∼0.6 (*Ahtiainen et al., 2016*; *Appleby, Newton & Cormie, 2012*; *Baker, Wilson & Carlyon, 1994*; *Balshaw et al., 2017*; *Cribb et al., 2007*; *Erskine, Fletcher & Folland, 2014*; *Erskine et al., 2010*; *Loenneke et al., 2017*; *Maeo et al., 2018*; *Pope et al., 2016*; *Rasch & Morehouse, 1957*; *Watanabe et al., 2018*). The discrepancies in findings between studies may be related, in part, to the statistical measures employed to analyze relationships between muscle hypertrophy and strength gain. For instance, analyses in a majority of studies are based on between-subject data using only two time points, but within-subject analyses are more appropriate for the question at hand. Inferentially, drawing individual-level conclusions from group-level data is a statistical fallacy, known as the ecological fallacy (*Robinson, 1950*). Pragmatically, this problem can be better understood by differentiating between the question that each analysis addresses. Between-subject analyses answer the question, "Do those who grow more also get stronger than those who grow less?" Conversely, within-subject analyses answer the question, "Is the growth of one's muscle related to their increases in strength?" Due to individual differences, the former (between-subject) may not necessarily map to the latter (within-subject). For example, if subject A has a 30% larger muscle moment arm than subject B, then one may expect subject A to have a 30% greater slope between increases in muscular strength (force) and externally-measured strength (moment), all else being equal. To address the ecological fallacy and answer the within-subject question, more sophisticated statistical approaches are needed (*Goldstein, 2011*; *Jackson, Best & Richardson, 2006*; *Robinson, 1950*).

A hierarchical approach can assist in avoiding the pitfall of the ecological fallacy (*Goldstein, 2011*; *Jackson, Best & Richardson, 2006*). Traditionally, each participant's change in strength and change in size, from pre- to post-intervention, are calculated and regressed among one another (*Ahtiainen et al., 2016*; *Appleby, Newton & Cormie, 2012*;

*Baker, Wilson & Carlyon, 1994*; *Balshaw et al., 2017*; *Cribb et al., 2007*; *Erskine, Fletcher & Folland, 2014*; *Erskine et al., 2010*; *Maeo et al., 2018*; *Pope et al., 2016*; *Rasch & Morehouse, 1957*; *Watanabe et al., 2018*). However, a hierarchical modeling approach allows for one to look at time points nested within participants, such that each participant's points are kept "separate" from other participants (*Gelman & Hill, 2007*; *Goldstein, 2011*; *Raudenbush & Bryk, 2002*). Within the hierarchical model, each participant can receive varying intercepts and/or varying slopes, which allows for inter-individual differences to be appropriately accounted for (*Gelman & Hill, 2007*; *Goldstein, 2011*; *Raudenbush & Bryk, 2002*). To carry out hierarchical modeling with varying slopes and intercepts, multiple ($\geq 3$) time points are required (i.e., to quantify model variance), so most training datasets cannot be used to answer this question, as a majority only collect data at two time points (pre- and post-intervention). To date, only one study has employed a within-subject analysis: *Loenneke et al. (2017)* used analysis of covariance (ANCOVA) (*Bland & Altman, 1995a*) and found appreciably greater coefficients of determination in within- relative to between-subject models for the same muscle and strength test (e.g., $R^2 = 0.004$ vs. $0.35$). However, in contrast to hierarchical linear models, ANCOVA has an affine assumption; participants receive different intercepts, but all are constrained to the same slope (*Bland & Altman, 1995a*). Therefore, further work is needed to understand how model choice affects the strength of the relationship between hypertrophy and changes in strength.

The purpose of this study was to investigate the relationship between changes in muscle size and strength in the elbow flexors using a variety of statistical and measurement approaches, while also employing both between- and within-subject analyses over multiple time-points during periods of both training and detraining. It was hypothesized that different statistical models would produce different outcomes, with between-subject correlations showing the weakest relationships and hierarchical linear modeling showing the strongest.

## METHODS

### Participants

The study reanalyzed data from a previously published study, the methods of which have been described (*Than et al., 2016*). In brief, young, recreationally active individuals (mean $\pm$ SD, age $= 24 \pm 3$ years, BMI $= 22 \pm 2$, $n = 19$) were recruited for participation in the study. Participants reported exercising at least three times per week via various sporting activities but did not perform resistance training for the elbow flexors. Informed consent was obtained for all participants. The original study was approved by the University of Queensland Medical Research Ethics Committee (no. 2014001416).

### Muscle size

Measures of muscle thickness were obtained via B-mode ultrasound imaging (Mindray DP-50) using a 7.5 MHz linear transducer probe. Images were taken at baseline and after each week of training throughout the 16-week study period. Scanning was carried out by a trained sonographer on both the dominant and non-dominant elbow flexors at 30, 50, and 70% of total length of the biceps brachii whilst participants were seated with the
antebrachium in a neutral position. After Weeks 4, 8, and 16, CSA scans were acquired for both upper limbs via panoramic B-mode ultrasound (S3000 Siemens/Acuson system) using a 4–9 MHz linear transducer operating at 9 MHz. Imaging for CSA was obtained via lateral acquisition at 50% width of the biceps brachii. Values for both muscle thickness and CSA were determined using ImageJ (version 1.48; National Institutes of Health, Bethesda, MD, USA). Muscle thickness was not assessed for Week 4 due to a conflict in scheduling with CSA ultrasounds. All ultrasound measures were completed by a paid qualified professional, and not by the researchers of the paper. If the probe lost contact at any point during the measurement, the measurement was retaken. Test-retest intraclass correlation coefficients (ICC; model 2,1) of 0.99 and 0.97 for CSA and muscle thickness, respectively, have been previously reported (*Jenkins et al., 2015*). Because an ICC(2,1) model was used, these results are generalizable to the experienced rater in this study (*Koo & Li, 2016*).

## Resistance training protocol

Resistance training for the non-dominant brachium was carried out five days per week for the initial eight weeks of the study, followed by a subsequent eight-week detraining period. Training consisted of unilateral dumbbell elbow flexion performed with a supinated forearm. During each session, participants performed nine sets of 12 repetitions with a 90-second rest interval afforded between sets. Loads were based on maximal voluntary isometric contraction (MVIC) values that were obtained each week using a Sundoo SN Analogue Force Gauge (model number SN-500) at 90° elbow flexion. Subjects began each workout using 70% of that week's MVIC recording. If the full number of target repetitions (i.e., 12) was not achieved on a given set, the load was lowered to the next level of load until completion—e.g., if a participant achieved 8 repetitions at 70%, the load was decreased to 50% so that all 12 repetitions could be performed. Loads were progressively lowered on successive sets to 50% and 30% of MVIC as needed so that subjects could complete the target repetition range with proper form. The dominant brachium of each subject served as the control for the study throughout the training and detraining periods. Subjects were instructed to refrain from exercise involving the elbow flexors, other than activities of daily living, throughout the 16-week study period.

## Statistical analysis

Several statistical analyses were carried out to investigate how methods of both measurement and analysis may affect the conclusions drawn from a study investigating the relationship between strength and hypertrophy. All analyses were carried out in R (version 3.4.3) (*R Core Development Team, 2017*). First, standard bivariate linear regression analyses of pre- and post-measures were utilized to investigate the relationship between muscle size (thickness or CSA) and strength, using a between-subject model. This was done for two different conditions: training and detraining. For each condition, a data point ($\Delta_{size}$, $\Delta_{strength}$) was calculated for each participant, where, in the general case, $\Delta = post - pre$, where pre and post are the values before and after a given condition (training or detraining), respectively, as has been done in a number of previous investigations (*Ahtiainen et al., 2016*; *Erskine, Fletcher & Folland, 2014*; *Loenneke et al., 2017*). Second, an ANCOVA was

utilized to replicate the method of analysis used by *Loenneke et al. (2017)*. In this analysis, strength was treated as a dependent variable, participants were treated as a categorical factor (dummy-coded), and size was treated as a covariate. Variance accounted for (VAF) was calculated using the formula $\text{VAF} = \frac{SS_{\text{size}}}{SS_{\text{size}} + SS_{\text{residual}}}$, where $SS$ is type III sum of squares (*Bland & Altman, 1995a*). This is equivalent to a partial $\eta^2$ for the size covariate. Lastly, because the ANCOVA method has a number of assumptions and does not allow for varying slopes, a more robust hierarchical linear model was used for the final analysis (*Quené & Van den Bergh, 2004*). In this analysis, the outcome measure ($y_{ij}$) was the net joint moment during MVIC, and muscle size was used as a level-one predictor variable ($x_{ij}$), which were group-mean centered for analyses. Subject was treated as a level-two variable. Finally, varied slopes and intercepts were permitted, creating the final model:

*Level* 1

$$y_{ij} = \beta_{0j} + \beta_{1j}x_{ij} + \epsilon_{ij}$$

*Level* 2

$$\beta_{0j} = \gamma_{00} + r_{0j}$$
$$\beta_{1j} = \gamma_{10} + r_{1j}$$

The model was fit using restricted maximum likelihood in the *lme4* package (*Bates et al., 2015*). Sample variance of the residuals ($s^2$) were used to calculate VAF (or $R^2$) using the following formula: $\text{VAF} = 1 - \frac{s^2}{s^2_{\text{uncond}}}$, where $s^2_{\text{uncond}}$ is the sample variance of the residuals in the unconditional model, which contained only varied intercepts and no fixed effects (i.e., the same model, but with $\beta_{1j} = 0$). This approach is mathematically equivalent to the VAF found for the ANCOVA using type III sums of squares (see Appendix A). Intraclass correlation coefficients (ICC) were calculated on the unconditional models to estimate the proportion of original variance explained by subject. To estimate 95% confidence intervals (CI) of the VAFs, each model was bootstrapped 2,000 times with replacement. The 0.025 and 0.975 quantiles of the VAF estimates were calculated as the lower and upper bounds of each estimate's 95% CI.

To understand how the different measures of hypertrophy relate to one another, within- and between-subject correlation matrices were constructed using the different thickness measures and CSA. The between-subject analysis included all thickness and CSA measures, across all subjects, for any time point at which both CSA and thickness were measured. The within-subject correlation matrix was constructed in a similar manner: (1) a correlation coefficient was calculated for each participant ($r_i$); (2) using a Fisher $z$-transformation, $r_i$ was transformed to a $z$-score ($z_i$); (3) a weighted average was obtained using the number of points ($n_i$) from each participant ($\bar{z} = \frac{\sum z_i(n_i - 3)}{\sum(n_i - 3)}$, for $i$ participants); and (4) $\bar{z}$ was transformed back to Pearson's $r$ (*Borenstein et al., 2009*; *Cooper, Hedges & Valentine, 2009*; *Corey, Dunlap & Burke, 1998*; *Hedges & Olkin, 1985*). Because CSA measures were only taken with thickness at two time points, within-subject correlation coefficients could not be estimated between CSA and muscle thickness.

**Table 1  Correlation coefficient and variance accounted for interpretations.**

| Interpretation | Correlation coefficient ($r$ or $\rho$) | Variance accounted for (%) |
| --- | --- | --- |
| Trivial | [0, 0.1) | [0, 1) |
| Small | [0.1, 0.3) | [1, 9) |
| Moderate | [0.3, 0.5) | [9, 25) |
| Large/strong | [0.5, 0.7) | [25, 49) |
| Very large/strong | [0.7, 0.9) | [49, 81) |
| Nearly perfect | [0.9, 1) | [81, 100) |
| Perfect | 1 | 100 |

Notes.
Adapted from *Hopkins (2002)*. Note that all intervals are of the form $x_{low} \leq x_o < x_{high}$.

Further exploratory analyses were performed to investigate if those with stronger strength-hypertrophy relationships also got stronger. To do this, Pearson correlation coefficients were calculated for each individual across the entire study (i.e., including both training and detraining periods). The resulting correlation coefficients were then correlated with $\Delta_{strength}$ from the training period using Spearman's rank-order correlations ($\rho$). Spearman's $\rho$ was used due to the heteroscedastic nature of the residuals. Qualitative interpretations of correlation coefficients and VAFs can be found in Table 1, which are in accordance with *Hopkins (2002)*. R code for all procedures can be found in the Supplemental Files.

# RESULTS

Differences in VAFs ranged from zero to an order of magnitude (Table 2). Similar differences were also observed between different statistical models for a given measure (Table 2). Intraclass correlation coefficients from the hierarchical linear models suggest that most of the original variance could be accounted for by including a level for subject (ICC = 0.89–0.91). Heterogeneity in correlation coefficients was observed when comparing different measures of muscle thickness, which ranged from $r = 0.503$ to $r = 0.945$ for between-subject correlations and from $r = 0.326$ to $r = 0.875$ for weighted within-subject correlations (Table 3). Finally, Pearson's $r$ of each individual's strength-hypertrophy relationship was a moderate to strong predictor of strength for all measurements (US$_{30\%}$ $\rho = 0.644$; US$_{50\%}$ $\rho = 0.356$; US$_{70\%}$ $\rho = 0.413$; US$_{avg}$ $\rho = 0.480$; CSA $\rho = 0.449$).

# DISCUSSION

To the authors' knowledge, this is the first study to investigate the relationship between hypertrophy and changes in muscle strength using hierarchical linear modeling, which allows for robust within-individual analysis, in addition to the use of multiple types of measures of muscle size. Our results demonstrate that not only does measurement approach substantially affect outcomes, but so does the type of statistical model employed. These findings have important methodological implications for improving our understanding of the associative relationship between hypertrophy and changes in strength.

**Table 2  Percent (%) variance accounted for (95% CI) using different types of models.**

| Measure | Between-subjects | | Within-subjects | |
|---|---|---|---|---|
| | **Training** | **Detraining** | **ANCOVA** | **HLM** |
| Thickness (30%) | 3.6 (0–61.9) | 1.0 (0–45.1) | 0.2 (0–6.1) | 7.4 (0.8–16.0) |
| Thickness (50%) | 0.8 (0–21.6) | 0.0 (0–23.7) | 0.3 (0–9.7) | 24.1 (6.7–42.0) |
| Thickness (70%) | 1.4 (0–39.1) | 1.6 (0–38.0) | 2.2 (0–10.9) | 7.5 (2.1–23.7) |
| Thickness (Average) | 0.4 (0–21.1) | 0.0 (0–26.4) | 1.2 (0–12.9) | 18.1 (6.6–30.4) |
| Cross-sectional area | 0.4 (0–32.2) | 1.2 (0–35.4) | 11.7 (1.1–34.2) | 12.1 (2.0–69.5) |

Notes.

30%, 50%, and 70% represent the position of the ultrasound probe on the brachium. Average represents the average of all three of the measured thicknesses at a given time point. Cross-sectional area was measured at 50%.
ANCOVA, analysis of covariance; HLM, hierarchical linear model.

**Table 3  Correlation matrix of measures of muscle size.**

| | **Thickness (30%)** | **Thickness (50%)** | **Thickness (70%)** | **Thickness (Average)** | **Cross-sectional area** |
|---|---|---|---|---|---|
| Thickness (30%) | | 0.503[a] | 0.618[a] | 0.778[a] | 0.557[a] |
| Thickness (50%) | 0.344[b] | | 0.869[a] | 0.916[a] | 0.742[a] |
| Thickness (70%) | 0.326[b] | 0.687[b] | | 0.945[a] | 0.730[a] |
| Thickness (Average) | 0.659[b] | 0.875[b] | 0.871[b] | | 0.773[a] |
| Cross-sectional area | | | | | |

Notes.

30%, 50%, and 70% represent the position of the ultrasound probe on the brachium. Average represents the average of all three of the measured thicknesses at a given time point. Cross-sectional area was measured at 50%.
[a] Between-subject correlation.
[b] Weighted within-subject correlation.

Previous literature has approached the question of how changes in muscle size relate to changes in strength from a between-subject perspective. However, it can be argued that a repeated-measures design allows for a more direct evaluation of the strength-hypertrophy relationship. Individual differences in muscle moment arms (MA), normalized muscle force (NMF), pennation angles ($\theta_p$), voluntary activation ($\alpha$), *et cetera* will greatly confound the relative relationship between changes in strength and muscle size (in this case, physiological CSA(PCSA)). All of the aforementioned components are multipliers in the formula used to calculate a muscle's contribution to a joint moment ($M = \alpha \cdot \text{PCSA} \cdot \text{NMF} \cdot \cos\theta_p \cdot \text{MA}$) (*Vigotsky, Contreras & Beardsley, 2015*). To date, only one previous investigation has utilized a quantitative within-subject approach to investigate the relationship between hypertrophy and changes in strength (*Loenneke et al., 2017*); although, qualitative within-subject changes are depicted in a classic study by *DeLorme (1945)*. Specifically, *Loenneke et al. (2017)* employed an ANCOVA with subject as a factor and muscle size as a covariate; from the resulting sum of squares, VAF could be calculated (*Bland & Altman, 1995a*). ANCOVA is limited, however, in that it, in its basic form, assumes parallelism between all relationships, has several assumptions that may confound results (e.g., sphericity, compound symmetry, and homoscedasticity), and is not robust to missing data points (*Bland & Altman, 1995a*; *Bland & Altman, 1995b*; *Quené & Van den Bergh, 2004*). The parallel or affine assumption is of particular interest because there are several heterogeneities

that confound this assumption (i.e., $\alpha$, MA, NMF, and $\theta_p$). Repeated-measures hierarchical models are a robust way to investigate longitudinal relationships within a group or person (*Gelman & Hill, 2007*; *Raudenbush & Bryk, 2002*). By comparing these statistical models, a clear difference is apparent (Table 2). For all measurements, the hierarchical linear model resulted in greater VAFs than the ANCOVA (Table 2). These differences may be due to the hierarchical linear model allowing for varying slopes or, alternatively, some of the inherent assumptions and limitations of ANCOVAs (*Quené & Van den Bergh, 2004*). Interestingly, the VAFs found in this present study are much lower than those found by *Loenneke et al. (2017)*. It is unclear from where these differences arise; that is, if they are due to measurement technique, differences in mechanisms of strength gain, differences in upper vs. lower extremities, or some other factor. However, our data provide a methodological proof of principle by delineating how different statistical models may drastically affect the conclusions formed from a given dataset, even when performed on the same set of regressors. Due to the robustness of hierarchical linear models, it is recommended that such analyses are used over ANCOVAs for future investigations with similar methods.

How muscle size is assessed will likely affect the strength of the relationship between changes in muscle size and strength. The measurement techniques utilized by previous and present investigations (*Ahtiainen et al., 2016*; *Appleby, Newton & Cormie, 2012*; *Baker, Wilson & Carlyon, 1994*; *Balshaw et al., 2017*; *Buckner et al., 2016a*; *Cribb et al., 2007*; *Erskine, Fletcher & Folland, 2014*; *Loenneke et al., 2017*; *Pope et al., 2016*) have been limited in that they do not account for changes in architectural characteristics (*Lieber & Ward, 2011*). There are several ways to measure muscle size, including limb circumference (*DeLorme, 1945*), estimates of total and segmental muscle mass (dual-energy X-ray absorptiometry and bioelectrical impedance analysis) (*Karelis et al., 2013*), muscle thickness (*Than et al., 2016*), anatomical CSA (*Erskine, Fletcher & Folland, 2014*; *Trezise, Collier & Blazevich, 2016*), muscle volume (*Balshaw et al., 2017*; *Erskine, Fletcher & Folland, 2014*; *Erskine et al., 2010*), and PCSA (*Erskine et al., 2010*). There are strong physiological and mechanical rationales with basic science evidence to suggest that not all of these measures are equal, even when accounting for measurement error (*Lieber & Ward, 2011*; *Powell et al., 1984*). For example, although muscle volume appears to be a strong predictor of strength in some contexts (even greater than anatomical CSA) (*Akagi et al., 2009*; *Fukunaga et al., 2001*), it does not perform as well in others (*Baxter & Piazza, 2014*), perhaps at least partly due to inter- and intra-muscular variation in architecture (*Blazevich, Gill & Zhou, 2006*; *Lieber & Ward, 2011*; *Ward et al., 2009*) and adaptation (*Earp et al., 2015*; *Ema et al., 2013*; *Franchi et al., 2017*; *Narici et al., 1996*; *Wakahara et al., 2013*; *Wakahara et al., 2012*). Muscle volume is not only sensitive to changes in sarcomeres in parallel (PCSA), but also sarcomeres in series (fiber length). Sarcomeres in parallel will contribute to the magnitude of force production, while sarcomeres in series will affect the shapes of the force-length and force-velocity curves. Functionally speaking, not all muscle volume is equal (*Lieber & Ward, 2011*). Importantly, in series hypertrophy appears to be limited to the initial weeks of commencing resistance training, further reinforcing potential issues when extrapolating correlative findings from novice to trained individuals (*Blazevich et al., 2007*). Similarly, thickness and anatomical CSA, as measured in this study, are also

limited, as they only represent one part of the muscle and do not account for the intricacies of muscle architecture. This is further evidenced by *Franchi et al. (2017)*, who found that, cross-sectionally, muscle thickness, anatomical CSA, and muscle volume are related, but the relative changes between muscle thickness and muscle volume did not strongly correlate following a training period. This is important when considering the formula for PCSA, in that the volume of the entire muscle must be taken into account (*Lieber & Ward, 2011*); not just thickness or anatomical CSA. Moreover, the variability in correlation coefficients between these measures may be a cause for concern (Table 3), in that it suggests not all measures of muscle size are necessarily capturing the same effects, which is elucidated further by the statistical models (Table 2).

Since PCSA has been shown to be a strong predictor of force production both *in vivo* (*Fukunaga et al., 1996*) and *in vitro* (*Powell et al., 1984*), it is considered the gold standard for relating muscle form (architecture) to function (force production) (*Lieber & Ward, 2011*). PCSA is, in essence, the "effective" CSA, as it is the average CSA perpendicular to the fibers' line of action. Thus, PCSA controls for pennation and is representative of the number of sarcomeres in parallel, making it highly indicative of a muscle's potential to generate force through the tendon (*Lieber & Ward, 2011*). It is imperative to consider these differences in measurement techniques in the context of this study and similar investigations (*Ahtiainen et al., 2016*; *Erskine, Fletcher & Folland, 2014*; *Erskine et al., 2010*; *Loenneke et al., 2017*). Although this study (Table 2) and others (*Loenneke et al., 2017*) have observed what is analogous to a strong correlation ($r \geq 0.5$) (*Hopkins, 2002*) with repeated-measures designs, substandard measurements of muscle size were used in the present study. Therefore, it is likely that PCSA measurements would produce different results (*Aagaard et al., 2001*). While PCSA is expensive to obtain and typically relies on MRI, newer technologies, such as 3D ultrasound, show promise as valid, affordable alternatives to MRI for estimating muscle volume and PCSA (*Barber, Barrett & Lichtwark, 2009*; *Barber et al., 2011*; *Haberfehlner et al., 2016*). Moving forward, it seems prudent that investigators utilize PCSA rather than other measures of muscle size, as the theory that hypertrophy leads to strength gains is predicated on this measure rather than other measures of muscle size.

The question of how changes in strength and changes in muscle size are related is one with broad clinical implications, ranging from the treatment and prevention of sarcopenia and dynapenia to exercise prescription for strength athletes. Clinically, if changes in muscle size are not important for strength, then exercise programs need not focus on variables that are more important for hypertrophy than strength, such as volume (*Ralston et al., 2017*; *Schoenfeld, Ogborn & Krieger, 2017*). Changes in strength do indeed arise from non-hypertrophic factors (*Folland & Williams, 2007*), including a myriad of neural adaptations (*Enoka, 1988*), in addition to changes in muscle moment arms (*Sugisaki et al., 2015*; *Vigotsky, Contreras & Beardsley, 2015*) and normalized muscle force production (*Erskine et al., 2010*), in which lateral force transmission has been suggested to play a role (*Jones, Rutherford & Parker, 1989*). This implies that changes in strength are interactive rather than linear. As such, how this relationship is investigated and modeled should reflect such complexities. First, with more reductionist strength testing (i.e.,

single-joint isometric testing), it can be argued that the ''skill'' component of strength is less relevant (as opposed to one-repetition maximum tests (*Buckner et al., 2016b*)), since little coordination is necessary and even untrained individuals see little-to-no changes in voluntary activation and co-contraction (*Behm, 1995*; *Erskine, Fletcher & Folland, 2014*; *Erskine et al., 2010*; *Noorkoiv, Nosaka & Blazevich, 2014*). Moreover, neural measures, such as voluntary activation, can be more accurately assessed during isometric efforts than during dynamic efforts (*Farina, 2006*; *Vigotsky et al., 2017*) and thus can more easily be incorporated into a final model. Second, measures of muscle size should reflect those in the model (i.e., using PCSA). While this is expensive and time consuming, it will provide more appropriate biomechanical insight (*Lieber & Ward, 2011*). Third, moment arm measures should be subject-specific and occur over the duration of an experiment, as moment arms may change with training (*Sugisaki et al., 2015*; *Vigotsky, Contreras & Beardsley, 2015*). Finally, longer duration studies may be more appropriate for several reasons: (1) individual response trajectories will vary, as evidenced by the high ICCs in this present investigation and the heterogeneous rank orders between time points in previous work (*Churchward-Venne et al., 2015*); (2) edema can greatly confound gross imaging measures of muscle size, depending on when the measurements are performed (*Damas et al., 2016*); (3) the magnitude of the difference between measurement points will be greater, which in turn will decrease the relative role of measurement error in parameter and VAF estimates (*Fuller, 1987*); and (4) to understand the extent to which contributions may or may not change over time. While this present study did not incorporate these recommendations, since it was based on previously collected data (*Than et al., 2016*), future studies should do so to properly isolate the associative contribution of muscle size (PCSA) to strength increases.

Thus far, our discussion has primarily focused on the associative, rather than causal, relationship between hypertrophy and strength gain. A conducive discussion of the causal nature of this relationship requires an operational definition of causality. In formal logic, causality is often broken down into two conditions: (1) necessary conditions, which state that *B* will not occur without *A* ("if not *A,* then not *B*"); and (2) sufficient conditions, which state that *A* will result in *B* ("if *A*, then *B*") (*Epp, 2011*; *Hall, 1987*). However, a less formal concept of causality is also possible without these conditions having been met, in the form of contributory causality. A contributory cause is neither necessary nor sufficient (*Hall, 1987*; *Riegelman, 1979*). Those who experience an effect need not experience its putative cause, and those who experience the putative cause need not experience its effect (*Riegelman, 1979*). For instance, although smoking causes lung cancer, not all of those who smoke develop lung cancer (i.e., it is not sufficient), and not all of those who develop lung cancer are smokers (i.e., it is not necessary); therefore, smoking may be viewed as a contributory cause of lung cancer (*Riegelman, 1979*). The arguments put forth by *Buckner et al. (2016a)*, *Dankel et al. (2018)* and *Mattocks et al. (2017)* do indeed rule out hypertrophy as being a necessary or sufficient cause for strength gain, but we suggest that the contributory nature of hypertrophy to strength should not be dismissed on this basis. In other words, changes in strength can occur without changes in muscle size and *vice versa*, but this does not preclude muscle size from contributing to strength. Experimentally, it is

PeerJ ______________________________________

important to consider the emergent, nonlinear, and interactive properties of strength; there are many moving parts that should be accounted for when attempting to understand such a complex system, which may concurrently change in different directions (e.g., increase in size but decrease agonist activation). Indeed, a systems rather than reductionist approach may be most appropriate for understanding strength emergence. In studying this system, it is necessary to measure all factors (confounders) that may contribute to strength to *truly* understand the role of hypertrophy, especially because different protocols may elicit differential adaptations (*Jenkins et al., 2017*). Thus, longitudinal, within-subject studies that incorporate all of the measures included in the formula to determine strength (PCSA, MA, activation and co-contraction, synergist characteristics, and NMF) are likely needed to better understand the emergent properties of strength. Finally, because the problem is so complex, the contributory role of hypertrophy in strength gain may not be able to be fully established from one study or line of evidence. Instead, a body of literature consisting of many forms of evidence—ranging from animal and agent-based models to observational and experimental human studies—may be required to elucidate the contributory role of hypertrophy in strength gain.

This study and its discussion have focused primarily on single muscle group hypertrophy and single-joint isometric strength gain. The larger question of multi-joint and dynamic strength gain is perhaps more relevant, but unfortunately much more complex (*Vigotsky et al., 2018*). Starting with relatively simpler systems and research questions may bear more fruit, while also providing a conceptual basis that can be used when studying more complex systems and research questions.

This is the first study to utilize repeated-measures hierarchical linear modeling to investigate the relationship between muscle size and strength. We herein demonstrate that repeated-measures hierarchical linear models produce different results than other within-subject models (ANCOVA), in addition to between-subject models, which is in line with previous work by *Loenneke et al. (2017)*. Moreover, it was found that different measures of muscle size can produce vastly different results. As such, we have advocated for more rigorous and reductionist experimental designs to better understand the mechanistic origins of single-joint strength following exercise programs, by suggesting that researchers measure PCSA and single-joint isometric strength, in addition to potential confounding variables.[1] These findings are important for the interpretation of previous studies, in addition to the design of future studies, on this same topic.

## CONCLUSIONS

The strength of the associational relationship between muscle hypertrophy and strength gain is highly dependent upon the statistical model employed. We have demonstrated that hierarchical linear modeling, which allows for varying slopes and intercepts, provides greater estimates of the strength of the relationship between muscle hypertrophy and strength gain. Moreover, different assessments of muscle size do not perfectly correlate, and therefore, different methods of assessment may lead to different conclusions. These findings should be taken into consideration when planning and interpreting studies on the relationship between muscle hypertrophy and strength gain.

[1] Note that these recommendations only apply to studies that are investigating the strength-hypertrophy relationship with a reductionist approach. We are in no way suggesting that PCSA and single-joint isometric measures be used for all resistance training studies.

### Funding

The authors received no funding for this work.

### Competing Interests

The authors declare there are no competing interests.

### Author Contributions

- Andrew D. Vigotsky conceived and designed the experiments, performed the experiments, analyzed the data, prepared figures and/or tables, authored or reviewed drafts of the paper, approved the final draft.
- Brad J. Schoenfeld conceived and designed the experiments, authored or reviewed drafts of the paper, approved the final draft.
- Christian Than and J. Mark Brown contributed reagents/materials/analysis tools, authored or reviewed drafts of the paper, approved the final draft.

### Human Ethics

The following information was supplied relating to ethical approvals (i.e., approving body and any reference numbers):

The original study was approved by the University of Queensland Medical Research Ethics Committee (no. 2014001416).

### Data Availability

The raw data and R analysis code are provided as Supplemental Files.

### Supplemental Information

Supplemental information for this article can be found online at http://dx.doi.org/10.7717/peerj.5071#supplemental-information.

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
