# Peer review of "Methods matter: the relationship between strength and hypertrophy depends on methods of measurement and analysis"

_PeerJ, doi:10.7717/peerj.5071_

## Round 0.1 · original submission · Minor Revisions

Thank you for a very well written submission. The reviewers have a few minor points and suggestions that I feel will strengthen the paper. Please address each point by both reviewers and be sure to highlight where you've made changes to the manuscript. If you disagree with any points made, feel free to provide a rebuttal. I look forward to the resubmission!
Scotty

·

Basic reporting

The paper is well written and follows a clear structure. Relevant literature is referred to with few relevant omissions. The author's clearly present their research question, and how it differs to those of previous investigations, and then offer a clear hypothesis for which the results specifically relate to.Relating to the basic reporting and writing of the manuscript I have only a few suggestions below in the General comments section.

Experimental design

The authors should be commended for the sharing of their analysis code alongside their data permitting others to replicate their study. There are some elements that I believe may need clarification in the methods.

1) Can you provide more detail on the CSA measures taken. I can imagine that taking clear biceps brachii CSA measures with ultrasound is difficult without a sufficiently wide linear probe and even then I often find that overlap with the contours of the surface of the upper limb means that contact with the probe can be lost. Was a panoramic image taken? I have checked the original manuscript and the details reported are the same? Further information would be appreciated.

2) Where any measures of the reliability of between day ultrasound measures determined?

3) How have you interpreted your statistical analysis? e.g. for correlations what was considered, weak, moderate, strong etc.

Validity of the findings

Results are presented clearly. Conclusions are stated clearly and relate to the original research question and hypothesis.

Additional comments

I think the authors should note and discuss the concerns of other authors around study design in determining 'cause and effect' relationships, as opposed to merely identify whether two things are related. The study of Mattocks et al.is discussed, but other papers highlighting and discussing this have not been referred to e.g. https://www.ncbi.nlm.nih.gov/pubmed/28819744

Line 53 - replace the comma after "...not limited to..." with a semi-colon.

Line 130 - change use of the term 'intensity' to 'load'

Line 208 - Add 'relationship' after 'strength-hypertrophy'

Lines 279-281 - I would be careful with statements such as this. The emergence of traits from an evolutionary perspective is determined by their 'net' effect on reproduction. Traits can emerge despite them having associated costs as long as their effect is a net positive upon reproductive success. These are often referred to as being 'costly' traits.

Line 295 - Although I agree with the statement that the time-course of adaptations likely varies between individuals, I'm not sure that the Churchward-Venne et al. paper supports this. There is not to my recollection data presented on an individual basis for any thing more than two time points - either 0 and 12wks, or 0 and 24wks.

Line 297 - I would suggest changing 'ensuring' to "...thus meaning that there is a greater chance that measurement error exceeded..."

·

Basic reporting

The Basic reporting of the manuscript was at a high standard, with appropriate references cited and the level of English professional and concise throughout.

Experimental design

I was impressed with most aspects of the experimental design; with these sections well described throughout.

Validity of the findings

The data presented in this manuscript is robust and clearly presents how a variety of data collection/analysis and/or statistical approaches can result in quite substantial differences in the results and ultimately the clinical interpretations of these results. Such findings have massive potential to influence our understanding of the relationship between hypertrophy and strength as well as the way that future researchers will need to examine this relationship in future studies.

Additional comments

General comments
This manuscript outlines a very well described rationale for how exercise and sport science researchers need to perhaps rethink our approaches to modelling the relationship between muscle mass and strength, be it in cross-sectional or training studies. I’m impressed with many aspects of this manuscript including the overall research question they sought to examine, the methods chosen to address this question and the way the manuscript is presented. A few small ways in which the manuscript can be improved are included below in the specific comments section.

Specific comments
line 45 – 46: please feel a bit more explicit and provide some examples of what some of these inter-individual differences may be.
Line 257: I think this sentence would be better written as “of the changes in these parameters did not correlate significantly over a training period”.
Line 271 – 273: you appear to have made a convincing argument for the importance of measuring PCSA, however this is not easily measured by most researchers and practitioners. Is there any possible way that this can be estimated based on other more easily obtained body composition measures?
Line 277 – 279: I think this sentence could be rephrased to improve its readability.
Line 290: I’m not sure what you mean here by the phrase “neural measures are more accurate than dynamic ones”. Please rephrase is to make it clear to the reader.
Line 294 – 298: this sentence is a little bit too long and challenging to read and I therefore suggest you break it into two or three smaller sentences to improve readability.
Line 313 – 325: one thought I had in relation to the challenging question of whether single joint or multi-joint strength exercises should be used in the context of the questions addressed in this paper may reflect the multifactorial determinants of strength across various joints and the roles of other muscles that act as synergist or stabilisers to the agonist muscles. As an example, could the results of a study in which you assess the hypertrophy of a variety of quadricep and gluteal agonist muscles and changes in squat 1RM be influenced by the potentially varying changes in hypertrophy and strength of muscles that contribute to trunk stiffness as these trunk muscles might be seen as the weak link in regards the potential to maximise the force production capacity of the lower limb in the squatting motion? As the hypertrophy and strength of these muscles influencing trunk stiffness are generally not measured in such studies, is this something that also warrants more discussion on this manuscript?
Overall: you have made a pretty strong case for the assessment of PCSA as the preferred measure of muscle hypertrophy, are you able to make such a case for the optimal method to measure muscle strength? Would this optimal measure differ for studies in which the relationship between strength and hypertrophy is the primary outcome compared to studies just concerned with improving strength?

---

## Round 0.2 · accepted · Accept

Congratulations on an excellent paper with very big implications for practice by a wide variety of audiences. Best wishes, Scotty

·

Basic reporting

Nothing additional to add.

Experimental design

The authors have added a reference indicating that ICC(2,1) was reported and that this supports the reliability of the rater in the study. While this is true, I personally would feel more comfortable with the authors at least noting that the lack of reliability data for the specific rater in this study is a potential limitation.

Validity of the findings

Nothing additional to add.

Additional comments

I am happy that the authors have adequately addresses my comments. This will be an interesting addition to the debate around this topic and I expect that it will likely stimulate further discussion and consideration of how to better answer the question of whether, and to what degree, changes in muscle size contribute to changes in strength. The authors included discussion regarding causality offers a balanced nuanced consideration of the topic.

·

Basic reporting

No comment

Experimental design

No comment

Validity of the findings

No comment

Additional comments

Congratulations on addressing my previous criticisms with the initial version of this manuscript. The only tiny revision I request is that you break a large paragraph into two smaller paragraphs. The paragraph I refer to starts on line 279 and finishes on line 329. This will help improve the readability of this section of the manuscript.